# Potential Anti-Diabetic Activity of *Pueraria lobata* Flower (Flos Puerariae) Extracts

**DOI:** 10.3390/molecules25173970

**Published:** 2020-08-31

**Authors:** Pattawika Lertpatipanpong, Sakawrat Janpaijit, Eul-Yong Park, Chong-Tai Kim, Seung Joon Baek

**Affiliations:** 1Laboratory of Signal transduction, Department of Veterinary Medicine, College of Veterinary Medicine and Research Institute for Veterinary Science, Seoul National University, Seoul 08826, Korea; pattawika.babeb@g.swu.ac.th (P.L.); Sakawrat.J@student.chula.ac.th (S.J.); 2Laboratory of Clinical Biochemistry and Molecular Medicine, Age-Related Inflammation and Degeneration Research Unit, Department of Clinical Chemistry, Faculty of Allied Health Sciences, Chulalongkorn University, Bangkok 10330, Thailand; 3R&D Center, EastHill Co. 33, Omokcheon-ro 132 beon-gil, Gwonseon-gu, Suwon-si, Gyeonggi-do 16642, Korea; rndhead@ieasthill.com (E.-Y.P.); ctkim@ieasthill.com (C.-T.K.)

**Keywords:** flos puerariae, PPARγ, GLUT4, antioxidant, iNOS, COX2

## Abstract

*Pueraria lobata* (Wild.) Ohwi. (*P. lobata*) flowers known as ‘Kudzu flower’ contain isoflavonoids and essential oil components. They have a wide range of biological and pharmacological activities, including protective effects against non-alcoholic fatty liver disease, hyperglycemia, and hypolipidemia, anti-mutagenic effects, and benefits for weight loss. However, the molecular mechanism of these effects remains unclear. Our study aimed to systematically examine the effects of flos puerariae crude extract (FPE) as an anti-diabetic agent using in vitro assays. The cytotoxicity of FPE was evaluated using MTS assay in L6 rat myocyte and 3T3-L1 murine fibroblast cell lines. PPARγ binding activity and adipogenesis were examined using dual-luciferase and differentiation assays, respectively. For investigating the anti-diabetic activity, glucose utilization, including GLUT4 protein expression, glucose uptake assay, and GLUT4 translocation using immunofluorescence microscopy were conducted in L6 cells. Furthermore, we assessed the antioxidant and anti-inflammatory activities of FPE. Our results demonstrated the ability to augment glucose uptake in L6 cells and enhance glucose utilization activity by increasing the expression of glucose transporter type 4 (GLUT4). In summary, our findings suggest that FPE may be a potential anti-diabetic substance for the treatment of diabetic patients and can prevent inflammatory or oxidation-related diseases.

## 1. Introduction

Diabetes mellitus is a metabolic condition with a rapidly increasing prevalence worldwide, currently affecting hundreds of millions of people. This multifactorial disease is characterized by hyperglycemia, insulin deficiency, insulin resistance, and/or abnormal insulin secretion. Diabetes can be classified into four categories: type 1 diabetes (T1DM), type 2 diabetes (T2DM), gestational mellitus (GDM), and other specific types of diabetes due to other causes. T2DM is characterized by impaired insulin-mediated glucose clearance in the skeletal muscle, but not always accompanied by dysregulation of hepatic glucose production by insulin. Insulin has dual roles in controlling postprandial glycemia response to a meal and activation of glucose transport into skeletal muscle and adipose tissue, both of which are impaired in T2DM [1].

Potential drugs for treatment of T2DM should preferably improve insulin resistance, correct dyslipidemia, and preserve pancreatic β-cell function. Regulation of gene expression in these pathophysiological conditions is activated by peroxisome proliferator-activated receptors (PPARs) that bind to either natural or synthetic ligands. Different subtypes of PPARs including peroxisome proliferator-activated receptor alpha (PPARα), PPARβ, and PPARγ have been identified [2]. Among these, PPARγ functions as a ligand-activated transcription factor and plays an important role in the regulation of adipogenic differentiation, glucose metabolism, insulin resistance, and inflammation [1,3].

Upon being bound by agonists, PPARγ heterodimerizes with retinoid X receptor-α and activates the transcription of target genes by binding to the peroxisome proliferator hormone response elements (PPRE) [3]. PPARγ is a key regulator of lipid and glucose metabolism and, therefore, its synthetic ligands such as glitazones, derivatives of thiazolidinediones (TZDs), improve insulin and glucose parameters and increase whole body insulin sensitivity. Thus, these are called insulin-sensitizing medications and are used in the treatment of T2DM. Synthetic drugs of the TZDs of anti-diabetic drugs efficiently bind to PPARγ; however, these treatments have various degrees of side effects. For instance, full agonists such as TZD drugs are clinically effective in treating T2DM; however, adverse drug effects such as fluid retention, edema, bone loss, liver toxicity, and weight gain have been reported. The mechanism underlying the anti-diabetic effects of TZD drugs also involve activation of cyclin-dependent kinase 5 (CDK5) by TZDs, resulting in the phosphorylation of PPARγ, leading to expression of specific genes linked to insulin sensitivity, such as adiponectin and adipsin [4]. Given the side effects of the current pharmacological treatments, it is necessary to identify safer agents for the treatment of DM, including remedies that may be more efficacious.

Traditional medicinal plants have an extensive history in the treatment of diseases and often offer promising leads for the treatment of T2DM. Increasing attention is being given to these resources for their complementary therapeutic effects that supplement western medicine. Notably, ethno-pharmacological studies have identified traditional remedies that might provide advantages in treatment with relatively fewer side effects, better patient tolerance in diverse cultural contexts, lesser toxicity, and lower costs, as compared to modern drugs.

*P. lobata* is a traditional medicinal plant that has been widely used in clinics for over thousands of years. Modern pharmacological studies have reported that *P. lobata* extracts have various therapeutic effects in cardiovascular disorders, osteoporosis, hypocholesterolemia, liver injury, and cancer, as well as other hyperglycemic defects [5,6,7]. It has been reported that the main compositions of flos puerariae are tectorigenin, saponins, and isoflavonoids [8,9,10].

In hyperglycemic studies, the root of *P. lobata* is capable of reducing the blood glucose levels in streptozotocin-induced diabetic mice, preventing pancreatic cells from the toxic action of reactive oxygen species [5,11]. However, the biological activity of *P. lobata* flowers has not been investigated in detail. Recently, a few studies indicated that flos puerariae has a protective effect against diseases, including non-alcoholic fatty liver disease [12], and also shows anti-mutagenic activity [13]. Furthermore, a single bioactive compound, *Puerarin,* isolated from *P. lobata* has been shown to prevent damage caused by free radicals and increase the activities of superoxide dismutase and catalase during oxidative stress [14]. Thus, it is important to explore how flos puerariae affects anti-diabetic signaling.

In this study, we identified the extract of *P. lobata* flowers as a strong PPARγ agonist among other tested plant extracts. We further found that these extracts exhibit anti-inflammatory and antioxidant activities in macrophages and favor glucose uptake activity in muscle cells. Thus, we postulate that flos puerariae may represent a beneficial therapeutic agent for T2DM.

## 2. Materials and Methods

### 2.1. Sample Preparation

All plants including flos puerariae were purchased from Kyung-Dong Market in Seoul, Korea. The authenticity had been confirmed at least twice through morphological and anatomical identifications by Dr. Jaeyoon Cha, Department of Food Science and Nutrition, Dong-A University, Busan, Republic of Korea. A voucher specimen (No. EHNP-F6) has been deposited in the R&D Center, EsatHill Corporation, Suwon, Gyeonggi-do 16642, Republic of Korea. The plants were washed and ground using a laboratory mill with a particle size of 100 mesh. 100 mL of ethanol (70%, Samchun Pure Chemical, Gyeonggi-do, Korea) was added to 10g of ground plants and extracted at 70 °C for 48 h with stirring at 500 rpm. The extracts were filtered using filter paper (Toyo No. 4, Advantec Toyo Kaisha, Tokyo, Japan) and concentrated using a vacuum evaporator (N-1110, EYELA, Tokyo, Japan). The extraction yields of *Puerariae lobata* flower (flos puerariae) extracts were approximately 9.5 ± 0.5%. Finally, the concentrate was diluted in dimethyl sulfoxide (DMSO, Biosesang, Gyeonggi-do, Korea) to obtain a final concentration of 100 mg/mL.

### 2.2. High Performance Liquid Chromatography (HPLC) Analysis of FPE

FPE extracts and standard compounds including Puerarin (98%, P1886, Tokyo Chemical Industry, Tokyo, Japan), Apigenin (98%, A1514, Tokyo Chemical Industry), Genistein (98%, G6776, Sigma-Aldrich, Saint Louis, MO, USA) and Rutin (95%, Zhejiang Medicines & Health Products Imp. & Exp Co, Hangzhou, China) were analyzed using Thermo Scientific™ Dionex™ UltiMate™ 3000 HPLC System with UV detector (Dionex, Sunnyvale, CA, USA). Gradient elution was carried out on Inno-C18 (250 mm × 4.6 mm, 5 μm) with the mobile phase consisted of CH3CN and 0.3% HCOOH aqueous solvent. The flow rate was 0.8 mL/min. Column temperature was held constant at 45 °C. The UV detection wavelength was 280 nm, 366 nm. Standard and sample (10 μL) were subjected to HPLC analysis.

### 2.3. Cell Culture

Mouse 3T3L1 pre-adipocytes and HCT116 human colorectal carcinoma cell lines were purchased from American Type Culture Collection (ATCC, Manassas, VA, USA) and Rat L6 myoblast and mouse Raw264.7 macrophage cell lines were purchased from Korean Cell Bank (Seoul, Korea). L6, Raw264.7, and 3T3L1 cells were grown in Dulbecco’s modified Eagle medium (DMEM, Gibco, Grand Island, NY, USA) supplemented with 100 U/mL penicillin and 100 mg/mL streptomycin (Gibco) at 37 °C in an incubator with a humidified atmosphere of 5% CO_2_ and for 3T3L1 cells, the cells were subsequently cultured in medium supplemented with 10% newborn calf serum (Gibco). HCT116 cells were grown in McCoy’s 5A Medium (Gibco, NY, USA) supplemented with 100 U/mL penicillin and 100 mg/mL streptomycin (Gibco) at 37 °C in an incubator with a humidified atmosphere of 5% CO_2_.

### 2.4. Cytotoxicity Assay

Cell cytotoxicity assay was performed using CellTiter 96^®^ AQueous One Solution Cell Proliferation Assay kit (Promega, Madison, WI, USA), as previously described [15].

### 2.5. Dual-Luciferase Assay

Luciferase reporter assay was performed using a Dual Luciferase Reporter assay kit (Promega), according to the manufacturer’s protocol. The vectors including p3XPPRE/tk/Luc (Addgene, Watertown, MA, USA) and pARE-luc (kindly provided by Dr. Young-Joon Surh of College of Pharmacy, Seoul National University, Seoul, Korea) were co-transfected with pRL-null using PolyJet^TM^ In Vitro DNA Transfection reagent (SignaGen, Frederick County, MD, USA). After transfection, cells were treated with various concentrations of FPE for 24 h. The relative luciferase activity was measured using Synergy^TM^ HTX multi-mode microplate reader (Biotek Instruments, Winooski, VT, USA).

### 2.6. T3-L1 Cell Differentiation

Plant extracts were added at concentrations of 10, 50, and 100 µg/mL to the 3T3-L1 cell cultured media throughout the adipogenic differentiation period. On the first day of differentiation, the cells were incubated with induction media containing 10% FBS, 0.5 mM IBMX (#I5879, Sigma-Aldrich, MO, USA), 0.25 µM dexamethasone (DEX, #D1756, Sigma-Aldrich), 2 µM rosiglitazone (#R2408, Sigma-Aldrich), and 1 µg/mL insulin (#I2643, Sigma-Aldrich). Following three days, the cell culture media were replaced with DMEM containing 10% FBS and 1 µg/mL insulin. Three days later, the cell culture media were changed to DMEM containing 10% FBS. The fully differentiated adipocytes were used for assay at 10 days after initiation of differentiation. The accumulated triglycerides were measured using Oil Red O assay. Briefly, after washing the cells twice with PBS, the cells were fixed using 4% paraformaldehyde (Duksan pure chemicals, Gyunggi-do, Korea) for 1 h at room temperature (RT). Following this, the cells were stained with 0.2% Oil Red O (Sigma-Aldrich) in 40% isopropranol (Daejung Chemicals, Gyeonggi-do, Korea) for 30 min and then washed five times with distilled water. Pictures of the stained cells were captured using a light microscope (Nikon Ti-U, Nikon Instruments, Tokyo, Japan). Oil Red O dye was eluded from the cell using 100% isopropanol and the optical density value was further measured at the absorbent wavelength at 500 nm using the Multiskan^TM^ FC microplate spectrophotometer (Thermo Fisher Scientific, Vantaa, Finland).

### 2.7. Immunofluorescence Analysis

L6 myoblast cell lines were transfected with GLUT4-EGFP plasmid using PolyJet^TM^ In Vitro DNA Transfection reagent (SignaGen, MD, USA). The transfected cells were treated with 10, 50, and 100 µg/mL FPE for 24 h. The cells were then washed twice with 1X PBS and fixed using 4% paraformaldehyde for 15 min at RT. DAPI (4′,6-diamidino-2-phenylindole, 1 µg/mL, Roche, IN, USA) was used for staining the nucleus of the cells for 15 min. Following this, the cells were washed five times with 1X PBS and mounted using Dako fluorescence mounting medium (#S3023, Dako North America, Carpinteria, CA, USA). The slides were then observed using a LSM900 confocal laser-scanning microscope (CLSM, ZEISS, Jena, Germany).

### 2.8. Glucose Uptake Assay

L6 myoblast cell lines were grown and differentiated by replacing the growth media with DMEM containing 2% FBS for five days before treating the cells with 10, 50, and 100 µg/mL FPE and 6 µg/mL insulin for 48 h. The glucose oxidase activity assay was performed in triplicate using the Glucose Oxidase Activity Colorimetric/Fluorometric assay kit (Biovision, Milpitas, CA, USA), according to the manufacturer’s protocol and measuring optical density at 500 nm using the MultiskanTM FC microplate photometer. The total consumption of glucose was calculated by subtracting the residue glucose in each sample well from the glucose concentrations of the blank wells.

### 2.9. Immunoblotting Analysis

Cells were washed twice in ice-cold PBS and lysed using a lysis buffer containing a proteinase inhibitor. The samples were sonicated, heated for 10 min at 95 °C, and centrifuged at 14,000× *g* for 20 min. Protein concentrations in the supernatants were determined using the Pierce^TM^ BCA protein assay kit (Thermo Scientific, Abbott Park, IL, USA). After the proteins were quantified, Western blotting was performed. Briefly, 60 µg proteins were separated using 10% sodium dodecyl sulfate-polyacrylamide gel electrophoresis and transferred to a nitrocellulose membrane. The blotted membrane was then blocked with 5% skim milk for 1 h at RT and incubated overnight with specific antibodies at 4 °C. The primary antibodies used (at a dilution of 1:1000) included mouse anti-GLUT4 (IF8, #sc53566 Santa Cruz, Starr County, TX, USA), rabbit anti-COX2 (D5H5, #12282, Cell signaling, MA, USA), rabbit anti-iNOS (D6B6S, #13120, Cell signaling), mouse anti-HO-1 (D8, #sc136961, Santa Cruz), rabbit anti-Cu/ZnSOD (#70R-CR010, Fitzgerald, Acton, MA, USA), and anti-β-actin (2A3, #sc517582, Santa Cruz). After incubation with goat anti-mouse IgG HRP-conjugated secondary antibody (1:5000 dilution, #62-6520, Thermo scientific) or goat anti-rabbit HRP-conjugated IgG secondary antibody (1:5000 dilution, #31460, Thermo scientific) in 5% skim milk for 1 h at RT, the blotted membranes were visualized using Alliance Q9 mini (Cambridge, UK) and quantified using ImageJ software 1.52a (National Institutes of Health, Bethesda, MD, USA).

### 2.10. Nitric Oxide Assay

Cells were seeded and grown until 70% confluency was reached, followed by treatment with various concentrations of FPE. Twenty-four hours after treatment, extract-treated media were replaced with serum-free DMEM containing 1 µg/mL lipopolysaccharide (LPS, #L4391, Sigma-Aldrich) and further incubated for 24 h. The media from each sample were collected and nitrite concentrations in the samples were determined using a nitric oxide colorimetric assay kit (Biovision, CA, USA), according to the manufacturer’s protocol by measuring the absorbance at 540 nm using the Multiskan^TM^ FC microplate photometer.

### 2.11. In Vitro Radical Scavenging Assay

DPPH (#14805, Cayman Chemical, Ann Arbor, MI, USA) and ABTS (#A1888, Sigma-Aldrich) were used for the radical scavenging assay as previously described [16]. The absorbance was measured using the Multiskan^TM^ FC microplate photometer. L-ascorbic acid (vitamin C) (#A0537, TCI, Tokyo, Japan) was used as a reference standard in both assays. Determination of percentage of radical scavenging effect was considered using the following equation: % Inhibition = 100 − [(Abs of sample − Abs of blank) × 100/Abs of control].

The vitamin C Equivalent Antioxidant Capacity (VCEAC) in mg/g dry weight extract and IC_50_ value, the half of the concentration of the sample that can scavenge 50% of the free radicals, were calculated.

### 2.12. Statistical Analysis

Data are expressed as mean ± SD or SEM from at least three independent experiments. Statistical analyses were performed using the one-way ANOVA test. All comparisons were made relative to untreated controls and significant differences have been indicated as * *p* < 0.05, ** *p* < 0.01, and *** *p* < 0.001. For screening of PPARγ ligand activity, analysis was performed using Dunnett’s multiple comparison test.

## 3. Results

### 3.1. Screening of PPARγ Ligand Activity Using Medicinal Plant Extracts

To identify plant extracts exhibiting PPARγ binding activity, the selected plants extracts were tested for their effects on the luciferase activity of a construct containing three copies of PPRE linked to luciferase (pPPRE3X/tk/Luc) (Figure 1A). HCT-116 cells were transfected with pPPRE3X/tk/Luc and then treated with these extracts. FPE showed the highest PPARγ binding activity among the tested extracts (Figure 1B). Subsequently, three different doses of FPE were tested for their PPARγ binding activity and it was found that FPE displayed PPARγ binding activity in a dose-dependent manner. Interestingly, cells treated with 100 μg/mL FPE exhibited higher luciferase activity, compared to 2 μM rosiglitazone used as a positive control.

### 3.2. Effect of Flos Puerariae Extract on 3T3-L1 Pre-Adipocytes

Firstly, we examined if FPE exhibits toxicity on the growth of 3T3-L1 preadipocytes. No cytotoxicity was observed upon maintaining 3T3-L1 cells in growth medium containing 100 μg/mL FPE (Figure 2A). Next, the effect of FPE on the adipogenesis of 3T3-L1 preadipocytes was determined by treating the cells with 10, 50, and 100 µg/mL FPE throughout the period of cell differentiation. All the concentrations of FPE displayed significantly enhanced adipogenesis of 3T3-L1 preadipocytes, as determined by the increasing size of droplets upon Oil Red O staining (Figure 2B). To quantitate the amount of lipid accumulation in each sample, we measured the OD of each treatment. The untreated 3T3-L1 preadipocytes showed an OD value of 0.323 ± 0.004, while the untreated mature adipocytes recorded an OD value of 0.408 ± 0.017. The ODs of the 10, 50, and 100 µg/mL FPE-treated cells were 0.497 ± 0.020, 0.511 ± 0.031, and 0.514 ± 0.022, respectively, which were significantly different from those of untreated 3T3-L1 preadipocytes and untreated mature adipocytes (Figure 2C). Higher OD values indicate enhanced differentiation of 3T3-L1 preadipocytes into mature adipocytes. Therefore, this result indicates that FPE increased the adipogenesis of 3T3-L1 preadipocytes in a dose-dependent manner, indirectly indicating that FPE works as a PPARγ ligand during preadipocyte differentiation.

### 3.3. Effect of Flos Puerariae Extract on Glucose Utilization

L6 cells were treated with FPE to determine its cytotoxicity on the cells. After treatment with 100 μg/mL FPE for up to three days, there was no significant difference with respect to cell growth, indicating that FPE is a safe extract for L6 cells (Figure 3A). The effects of FPE on the glucose utilizing activity of L6 myocytes were determined by assaying GLUT4 protein expression, GLUT4 trafficking, and glucose uptake activity. To measure GLUT4 protein expression, we performed an immunoblotting assay and found that the relative ratio of GLUT4 protein expression as compared to untreated L6 myocytes increased in a dose-dependent manner with increasing concentrations of FPE (Figure 3B–C). Upon measuring glucose uptake, the relative ratios in cells treated with 6 µg/mL insulin were found to be 2.545 ± 0.036-fold as compared to untreated L6 myocytes. In case of FPE treatment, the relative ratios were found to be 1.281 ± 0.210, 1.695 ± 0.175, and 1.740 ± 0.129-fold in samples treated with 10, 50, and 100 µg/mL FPE, respectively, as compared to the untreated cells (Figure 3D). Immunofluorescence was conducted using a fluorescence microscope to investigate GLUT4 protein trafficking. We found that L6 cells co-treatment with 100 nM insulin and 100 µg/mL FPE increased the amount of translocated protein from the cytoplasm to the membrane, as compared to untreated L6 myocytes (Figure 3E). Our results indicate that FPE had the ability to affect glucose trafficking in L6 myocytes in a dose-dependent manner.

### 3.4. Antioxidant Activity of Flos Puerariae Extracts

DPPH and ABTS radical scavenging activities were measured to determine the antioxidant capacity of FPE at different concentrations. The ability to reduce ABTS was found to be higher at the higher concentrations of FPE (Figure 4A). The VCEAC of FPE was found to be 179.639 ± 16.151 mg/g dry weight, whereas percentages of ABTS scavenging activity were 38.421 ± 1.708, 58.865 ± 1.299, 77.112 ± 0.452, 88.129 ± 0.358, and 93.288 ± 0.066 for 62.5, 125, 250, 500, and 1000 µg/mL FPE, respectively. In the case of DPPH radical scavenging activity, we observed a similar result, i.e., the IC_50_ of FPE was 882.267 ± 52.726 µg/mL and the percentages of DPPH scavenging activity were 9.361 ± 2.000, 16.052 ± 3.389, 28.618 ± 6.374, 48.905 ± 8.638, and 57.363 ± 9.587 for 62.5, 125, 250, 500, and 1000 µg/mL FPE, respectively (Figure 4B). To confirm FPE’s antioxidant activity, a luciferase construct containing antioxidant response element (ARE) was transfected into Raw264.7 cell lines. FPE treatment in the transfected cells resulted in increased luciferase activity, indicating that FPE may activate NRF2 protein followed by activation of ARE promoter (Figure 4C). Trans-Chalcone (25 μM) was used as a positive control for the antioxidant activity. The calculated relative ratio of ARE-luciferase activity was 5.652 ± 0.299-fold for positive control and 2.049 ± 0.139, 2.152 ± 0.173, and 2.877 ± 0.137-fold for 10, 50, and 100 μg/mL FPE-treated cells, respectively, compared to untreated cells (Figure 4C).

### 3.5. Anti-Inflammatory Activity of Flos Puerariae Extracts

To assess the anti-inflammatory effects of FPE, we conducted several assays, including immunoblotting and nitric oxide assay. Upon performing immunoblotting, we found that the expression levels of iNOS and COX-2 increased following treatment with 1 µg/mL LPS; however, FPE treatment was found to abolish the LPS-induced increase in iNOS and COX2 protein expression. These effects were dose-dependent, as shown by treatment with 10, 50, and 100 µg/mL FPE (Figure 5A). To determine the nitrate concentration, we performed a nitric oxide assay using RAW264.7 cells. LPS treated-cells displayed a 1.744 ± 0.094-fold increase in the nitrate levels; however, the amount of nitrate was found to be diminished upon FPE treatment (Figure 5B). Finally, the HO-1 expression level was measured in RAW264.7 cells, following FPE treatment. FPE increases HO-1 expression in a dose-dependent manner, but not Cu/ZnSOD, indicating that FPE may have a unique pathway to enhance antioxidant activity (Figure 5C). Finally, we performed the HPLC method to detect bioactive compounds in FPE. Four bioactive compounds, probably existing in FPE were examined (Figure 6A). Although no detectable amounts of Peurarin (major of compound in P. lobate root) and Rutin were identified, we found the FPE to contain apigenin and genistein, which may play roles in anti-oxidant, anti-inflammatory, and anti-diabetic activities.

## 4. Discussion

Numerous potential plant extracts have been used for the treatment/prevention of various diseases, including DM. PPARγ acts as a receptor for molecules with anti-diabetic properties and is also capable of modulating adipogenesis; thus, plant extracts as well as single compounds are assayed for their PPARγ binding activity to screen those with a potential role in controlling diabetes. The present study demonstrates that the extract from flos puerariae exhibits in vitro anti-diabetic activity in a dose-dependent manner, as assessed by PPARγ binding activity assay, glucose utilization in L6 myocytes, anti-inflammatory and antioxidant activities, and enhancement of adipogenesis by accumulating lipid droplets in 3T3-L1 cells.

The current available treatments for T2DM act through diverse mechanisms to improve glycemia. Many of these treatments also exert anti-inflammatory and/or antioxidant effects that might be mediated via their metabolic effects on hyperglycemia and hyperlipidemia or by directly modulating the response of immune system. Part of these findings regarding the effects of different medications on systemic and tissue-specific inflammation have been obtained using in vitro or in vivo models.

In mammalian cells, the facilitative diffusion of glucose across the plasma membrane is mediated by a family of glucose transporters. GLUT4, one of the glucose transporter isoforms, is only expressed in peripheral tissues that are targets for insulin action, including adipose tissue, cardiac muscle, and skeletal muscle. In in vivo models, transgenic mice that express high GLUT4 level in adipose tissue [17] or in skeleton muscle [18] are both highly insulin sensitivity and glucose tolerance. In contrast, mice with ablation of GLUT4, specifically in the muscle, show the expression of insulin resistance and diabetes from a young age. Originally, insulin resistance occurs in the skeleton muscle; however, later on, these mice become insulin resistant not only in the adipose tissue but also the liver [19,20]. Upon acute stimulation by insulin, GLUT4 transporters translocate from their intracellular compartment to the plasma membrane and, therefore, are responsible for insulin-stimulated glucose uptake. In the present study, glucose uptake was seen upon treatment of L6 cells with FPE; this is probably a result of GLUT4 translocation as shown by immunofluorescence (Figure 3E). FPE increased the expression levels of GLUT4 as well (Figure 3B). Overall, FPE shows anti-diabetic activity by increasing GLUT4 expression, followed by enhancing translocation as well as glucose uptake.

Chronic inflammation contributes to the development of many diseases, including cancer, arthritis, heart disease, and diabetes and its complications. Cyclooxygenase-2 (COX-2) is a key molecule that increases the inflammatory status of cells leading to pathophysiological conditions by increasing prostaglandin formation. It also leads to excessive formation of reactive oxygen species [21]. Persistent hyperglycemia and a chronic inflammatory environment lead to oxidation-mediated stress and injury and also alter insulin sensitivity by triggering different key steps in the insulin-signaling pathway [22]. Cyclooxygenase-2 inhibitors, members of the non-steroidal anti-inflammatory drug class, have been shown the ability to alleviate the augmented inflammatory responses in several models of diabetes [23,24,25]. However, their adverse effects on targets, including impaired renal perfusion and its function, peptic ulcers, and enhanced risk of myocardial infarction, contribute to limited use of these drugs for clinical prevention/treatment of T2DM [23]. Furthermore, several studies on T1DM and T2DM report increased formation of free radicals and decreased antioxidant capacity in the disease condition, which contribute to oxidative stress that damages cells [26]. Thus, anti-inflammation and antioxidant activities of any extract or compound can be beneficial for diabetic patients. In addition to COX-2 and iNOS protein expression, we observed that FPE induced the expression of HO-1 protein. HO-1, mostly studied for its cytoprotective properties in conditions associated with inflammation and oxidative stress, is an inducible rate-limiting enzyme that catalyzes free heme into carbon monoxide, free iron, and biliverdin [27,28,29,30]. It has also been reported that dysregulation of the HO-1 system is associated with several inflammatory disorders, such as atherosclerosis and rheumatoid arthritis [27], suggesting that HO-1 is also a potent anti-inflammatory mediator and shows the potential to be a molecular target for FPE in mediating its anti-inflammatory activity. Treatment with LPS has been shown to induce the production of NO in macrophage cells (Figure 5A) as well as adipocyte cells [31]. To investigate whether FPE alleviates insulin resistance by inhibiting NO production, we measured nitrite concentration in Raw264.7 cells and found FPE to be capable of decreasing NO production in macrophages (Figure 5B). These results suggest that FPE reduces insulin resistance, probably through reduction in NO levels, which could alleviate the impairment in the insulin-signaling cascade. Thus, it indirectly exhibits potential properties that could prevent diabetic complications caused by inflammatory and oxidative stress.

## 5. Conclusions

There are various molecular targets for T2DM treatments such as the IRS1/PI3K/Akt pathway, AMPK pathway, and the GLP-1 pathway. Our study showed that FPE activates the PPARγ nuclear receptor as a ligand and exhibits anti-inflammatory and antioxidant activities upon treatment in cells. It also affects glucose metabolism by increasing GLUT4 expression (Figure 6B). Overall, our results indicate that FPE displays potential anti-diabetic activity; however, further analysis of this extract is required to elucidate the detailed molecular mechanisms of this activity.

## Figures and Tables

**Figure 1 molecules-25-03970-f001:**
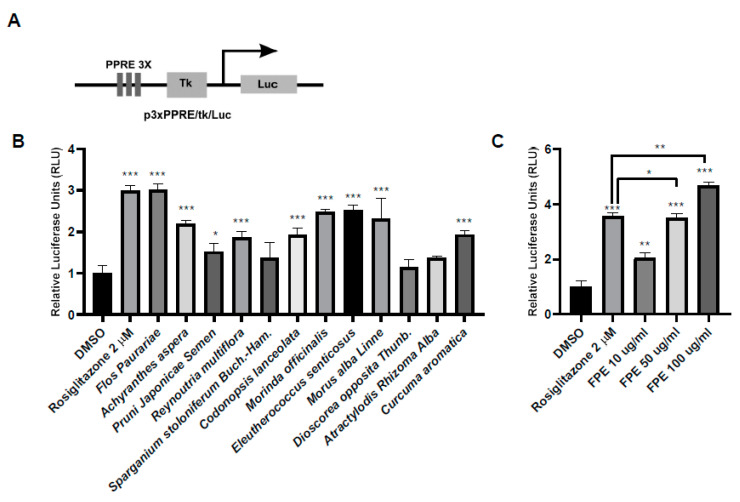
FPE induces PPARγ activity, as assessed by dual-luciferase assay. (**A**) Structural elements of the PPARγ luciferase reporter vector. (**B**) Screening of PPARγ activity in transiently transfected with pcDNA3/PPARγ, p3xPPRE/tk/Luc, and pRL plasmids into HCT116 cells and subsequently treated with 50 µg/mL plant extracts for 24 h. The dual-luciferase assay was performed. The relative values are presented as mean ± SEM; *n* = 3, * *p* < 0.05, ** *p* < 0.001, *** *p* < 0.0001. (**C**) Ability of FPE to induce PPRE-luc expression, PPARγ ligand-dependence, showed as dose-dependent manner. HCT116 cells were transiently transfected with pcDNA3/PPARγ, p3xPPRE/tk/Luc, and PRL null plasmids were treated with 10, 50, and 100 µg/mL FPE. Cells treated with 2 µM rosiglitazone were considered as a positive control. Dual-luciferase assay was performed 24 h after treatment of cells with the plant extracts. Results were statistically analyzed using Dunnett’s multiple comparison test; *n* = 3, * *p* < 0.05, ** *p* < 0.01, *** *p* < 0.0001.

**Figure 2 molecules-25-03970-f002:**
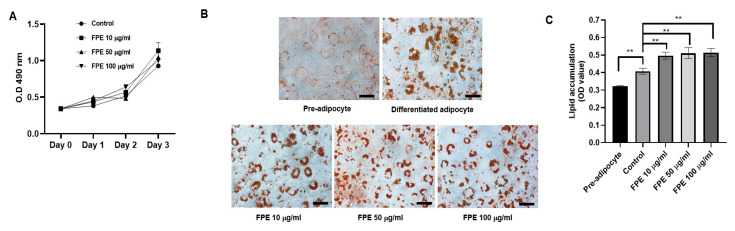
Effect of FPE on 3T3L1 pre-adipocytes. (**A**) Cytotoxicity assay of 3T3L1 cell lines treated with various concentrations (10, 50, and 100 µg/mL) of FPE displayed no significant toxicity effect on 3T3L1 pre-adipocytes at 0, 24, 48, and 72 h post-treatment with FPE. (**B**) Effect of FPE in enhancing adipogenesis in pre-adipocyte cell lines. 3T3-L1 preadipocytes were differentiated to adipocytes; the cells were treated with 10, 50, and 100 µg/mL FPE throughout the period of differentiation. At day 10 after initiation of differentiation, cells were stained with Oil Red O and visualized using a light microscope. Each figure shows the amount and size of droplets of adipocyte cells upon Oil Red O staining. Scale bar = 50 µm. (**C**) Adjacent panel graphically shows the absorbance of the lipid droplets after staining cells with Oil Red O for 30 min. Results are given as mean ± SEM; *n* = 3, ** *p* < 0.01.

**Figure 3 molecules-25-03970-f003:**
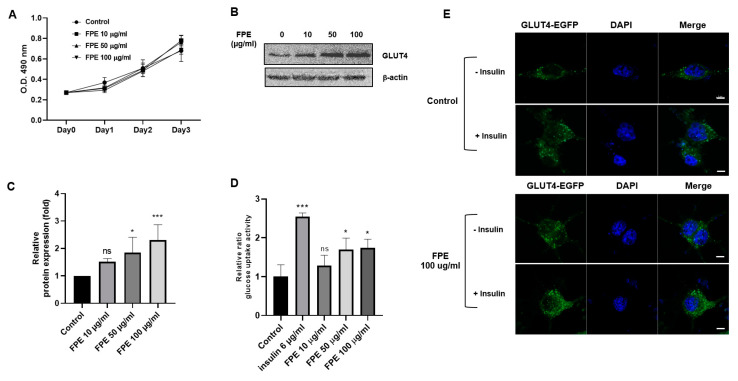
Effect of FPE on L6 myocytes. (**A**) L6 cells treated with 10, 50, and 100 µg/mL FPE showed no significant toxicity at 0, 24, 48, and 72 h post treatment. (**B**) Western blotting showed a dose-dependent increase in GLUT4 protein expression in rat L6 myocytes treated with increasing concentrations of FPE for 24 h. (**C**) The bar graphs represent the relative protein expression levels of GLUT4 after normalization to β-actin. Results are shown as mean ± SEM; *n* = 3, ns, not significant; * *p* < 0.05, *** *p* < 0.001. (**D**) Glucose oxidase assay was performed to quantitate the amount of glucose consumption in differentiated L6 myocytes treated with 6 µg/mL insulin and 10, 50, and 100 µg/mL FPE. The results indicated that the glucose uptake ability was significantly higher in cells treated with 50 and 100 µg/mL FPE but not significantly different in cells treated with 10 µg/mL FPE compared to untreated cells. *n* = 3, * *p* < 0.05, *** *p* < 0.001. (**E**) Effects of FPE on GLUT4 trafficking, compared to negative control and insulin as positive control. GLUT4-EGFP plasmid (green) was transiently transfected into L6 myocytes and co-treated with 100 nM insulin and 100 µg/mL FPE for 24 h. The cells were then fixed with 4% paraformaldehyde and the nuclei (blue) were stained with 1 µg/mL DAPI and examined using a confocal microscope. Scale bar = 5 µm.

**Figure 4 molecules-25-03970-f004:**
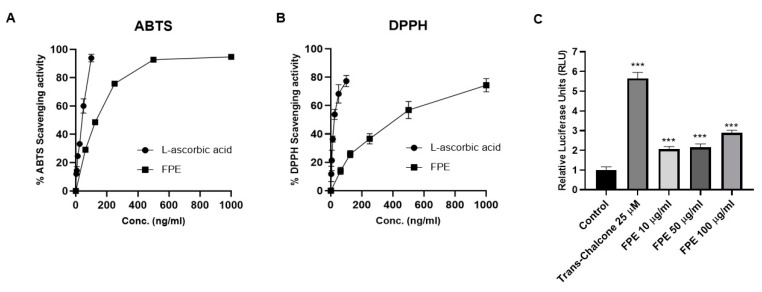
Antioxidant activity of FPE measured by biochemical assays. (**A**) 2,2-Azino-bis3-ethylbenthiazoline-6-sulfonic acid (ABTS) radical scavenging ability of the 95% ethanol fraction of flos puerariae. ABTS assay showed a dose-dependent enhancement in the antioxidant activity of FPE. Quantitation of the results from three independent experiments (*n* = 3) is shown as mean ± SD with statistical significance as * *p* < 0.05, ** *p* < 0.01, and *** *p* < 0.001. (**B**) The DPPH radical scavenging activities of the 95% ethanol fraction of flos puerariae. The result showed a dose-dependent enhancement in the antioxidant activity of FPE. Quantitation of the results from three independent experiments (*n* = 3) is shown as mean ± SD with statistical significance as * *p* < 0.05, ** *p* < 0.01, and *** *p* < 0.001. (**C**) Dose-dependent effect of FPE in inducing ARE-luciferase expression. HCT116 cells transiently transfected with ARE-luciferase plasmid vectors were treated with 10, 50, and 100 µg/mL FPE. Cells treated with 50 µg/mL trans-Chalcone were considered as a positive control. Dual-luciferase assay was performed 24 h after treatment of cells with plant extracts. Results are expressed as mean ± SEM; *n* = 3, *** *p* < 0.0001.

**Figure 5 molecules-25-03970-f005:**
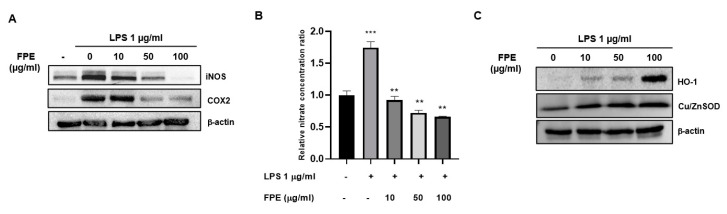
The anti-inflammatory activity of FPE in Raw264.7 murine macrophages was found to increase in a dose-dependent manner. (**A**) Western blotting showed a dose-dependent effect on the expression levels of iNOS and COX2 proteins following co-treatment of cells with 1 µg/mL LPS and 10, 50, and 100 µg/mL FPE for 24 h. (**B**) The adjacent bar graphs demonstrate the dose-dependent capacity of FPE to lower nitrate levels in Raw264.7 murine macrophages, following co-treatment with 1 µg/mL LPS and 10, 50, and 100 µg/mL FPE. Results are shown as mean ± SEM; *n* = 3, ** *p* < 0.01, *** *p* < 0.0001. (**C**) Western blotting of HO-1 and Cu/ZnSOD protein expression. The result shows expression levels of both proteins when cells were treated with 10, 50, and 100 µg/mL FPE for 24 h in a dose-dependent manner.

**Figure 6 molecules-25-03970-f006:**
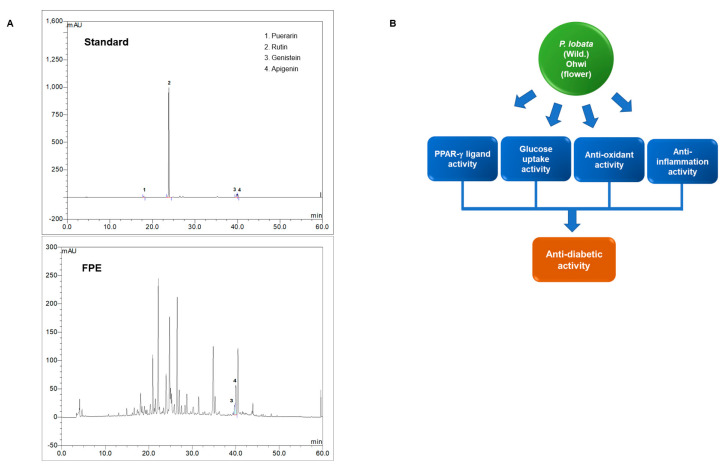
(**A**) Identification of 4 possible active compounds in FPE by HPLC. Peak identification is as follows: 1, Puerarin (retention time 18.00 min); 2, Rutin (retention time 23.84 min); 3, Genistein (retention time 39.81 min); 4, Apigenin (retention time 40.07 min). (**B**) Schematic diagram showing the anti-diabetic properties of FPE, including PPARγ ligand binding, glucose uptake, anti-inflammatory, and anti-oxidant activities.

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
