# Peer review of "Potential Anti-Diabetic Activity of *Pueraria lobata* Flower (Flos Puerariae) Extracts"

_molecules, 2020, doi:10.3390/molecules25173970_

Round 1
Reviewer 1 Report
The article by Lertpatipanpong et al. aimed at the investigation cytotoxicity of Pueraria lobata extract using MTS assay in L6 rat myocyte and 3T3-L1 murine fibroblast cell lines, PPARγ binding activity, GLUT4 protein expression, and antioxidant potential. I have no comments about biological methods and results. There are executed at an appropriate level. But I was confused by the absence of any chemical data. You have to characterize your extract particularly with regard to the content of the marker compounds (puerarin, daidzin, daidzein, and rutin). The adequacy of biological data cannot be determined without knowledge of this data.
Also some minor remarks:
- sect. 2.1: the yield of dry extract should be indicated;
- the reference marks should be like this [4], not like that 4;
- reference sect.: the rules of Reference section presentation described in some details. Please check this section precisely. Compare: 1. Janani, C.; Ranjitha Kumari, B. D., PPAR gamma gene – A review. Diabetes Metab Syndr. 2015, 9 (1), 46-50. and 1. Janani, C.; Ranjitha Kumari, B. D., PPAR gamma gene – A review. Diabetes Metab. Syndr. 2015, 9, 46-50. doi:10.1016/j.dsx.2014.09.015
- ref. 21: Donath, M. Y., Targeting inflammation in the treatment of type 2 diabetes: time to start. - what does it mean?
- ref. 22: Nat. Rev. Drug discovery 2014, 13 (6), 465-76. - the same question.
- Where are the doi? "Include the digital object identifier (DOI) for all references where available." - see line 138 of the Molecule template.
- Sample Availability: Samples of the compounds ...... are available from the authors. What kind of compounds did you mean?
Author Response
We would like to thank both reviewers very much for their thoughtful reading of our manuscript and constructive criticism. We have made additions to the text to address comments and concerns. An attached file contains the reviewer comments in bold, followed by our responses in red.

Reviewer 2 Report
This work focuses on the determination of Puerariae lobata flower's anti-diabetic activity. Research in this form has not yet been conducted. The obtained research results may expand the current state of knowledge on the activity of flowers. So far, research has been carried out using the root. The work is interesting and well described but requires some additions and improvements. I propose to accept the work after minor revisions.
ABSTRACT:
The abstract is properly described.
"It has a wide range of biological and pharmacological activities, including protective effect in non-alcoholic fatty liver disease, hypoglycemia..."
- "hypoglycemia" or "hyperglycaemia"?
INTRODUCTION:
The introduction contains relevant information that confirms the validity of this research.
The abstract mentions the chemical composition of Kudzu flower. This information is missing in the introduction. It is worth writing a few words about the presence of active compounds. Are isoflavones responsible for the potential health-promoting effect?
"postprandial glycemiaresponse"
- space between "glycemia" and "response"
"P. lobata is a traditional medicinal plant that has been wildly used"
- "wildly" or "widely"?
MATERIALS AND METHODS
Please provide detailed data (model, manufacturer, country) of the equipment used (e.g. microplate photometer, vacuum evaporator, water bath?) where such information is missing.
Please provide the manufacturer of reagents used (e.g. ethanol, dimethyl sulfoxide).
What were the sample weight and the volume of solvent used during the extraction?
The individual tests were properly described.
RESULTS
The results were presented in an understandable and appropriate manner.
I propose to divide the presented figures. Figures 3 and 4, in particular, have a very long description, which in my opinion may make it difficult for the reader to understand. However, the descriptions provided were prepared in an appropriate manner.
DISCUSSION
This section is described correctly.
CONCLUSIONS
"however, further analysis of this extract is required to elucidate the detailed molecular mechanisms of this activity"
- Can the authors suggest what specific studies should be performed to improve the current state of knowledge?
Kudzu root is commonly sold as a dietary supplement. Does Kudzu flower have any status related to human consumption? Can you conclude on its safety of consumption, have any studies been done on its side effects? What is the current state of knowledge related to safe use? It is worth briefly writing about it to indicate the potential possibilities and limitations of introducing Kudzu flower for human use.
REFERENCES
Please verify the literature.
Journal abbreviations require a "dot"
There should be a "dot" between the last name and the title of the article.
Author Response

(The authors gave the same response as above.)

Round 2
Reviewer 1 Report
The authors took all comments into account. The final recommendation: Accept in the present form.
Author Response
As the editor suggested, we included HPLC data as a new figure (Figure 6A). We also included method, result, and figure legend accordingly. We really appreciated the editor's effort.